# PDK1 Inhibitor BX795 Improves Cisplatin and Radio-Efficacy in Oral Squamous Cell Carcinoma by Downregulating the PDK1/CD47/Akt-Mediated Glycolysis Signaling Pathway

**DOI:** 10.3390/ijms222111492

**Published:** 2021-10-25

**Authors:** Shin Pai, Vijesh Kumar Yadav, Kuang-Tai Kuo, Narpati Wesa Pikatan, Chun-Shu Lin, Ming-Hsien Chien, Wei-Hwa Lee, Michael Hsiao, Shao-Chih Chiu, Chi-Tai Yeh, Jo-Ting Tsai

**Affiliations:** 1Graduate Institute of Clinical Medicine, College of Medicine, Taipei Medical University, Taipei 110, Taiwan; einfachweiss@gmail.com (S.P.); mhchien1976@gmail.com (M.-H.C.); 2Department of Oral & Maxillofacial Surgery, Saint Martin de Parres Hospital, Chiayi City 600, Taiwan; 3Department of Medical Research and Education, Taipei Medical University-Shuang Ho Hospital, New Taipei City 235, Taiwan; vijeshp2@gmail.com (V.K.Y.); narpatiwp@gmail.com (N.W.P.); whlpath97616@s.tmu.edu.tw (W.-H.L.); 4Department of Medical Laboratory Science and Biotechnology, Yuanpei University of Medical Technology, Hsinchu City 300, Taiwan; 5Division of Thoracic Surgery, Department of Surgery, School of Medicine, College of Medicine, Taipei Medical University, Taipei 110, Taiwan; doc2738h@gmail.com; 6Division of Thoracic Surgery, Department of Surgery, Shuang Ho Hospital, Taipei Medical University, New Taipei City 235, Taiwan; 7Department of Radiation Oncology, Tri-Service General Hospital, National Defense Medical Center, Taipei 114, Taiwan; chunshulin@gmail.com; 8Department of Pathology, Taipei Medical University-Shuang Ho Hospital, New Taipei City 235, Taiwan; 9Genomics Research Center, Academia Sinica, Taipei 115, Taiwan; mhsiao@gate.sinica.edu.tw; 10Graduate Institute of Biomedical Sciences, China Medical University, Taichung 404, Taiwan; scchiu@mail.cmu.edu.tw; 11Translational Cell Therapy Center, Department of Medical Research, China Medical University Hospital, Taichung 404, Taiwan; 12Drug Development Center, China Medical University, Taichung 404, Taiwan; 13Department of Radiology, School of Medicine, College of Medicine, Taipei Medical University, Taipei 110, Taiwan; 14Department of Radiology, Taipei Medical University-Shuang Ho Hospital, New Taipei City 235, Taiwan

**Keywords:** OSCC, PDK1, CD47, cancer stem cells, glycolysis, chemo/radioresistance

## Abstract

**Background**: Oral squamous cell carcinoma (OSCC) has a high prevalence and predicted global mortality rate of 67.1%, necessitating better therapeutic strategies. Moreover, the recurrence and resistance of OSCC after chemo/radioresistance remains a major bottleneck for its effective treatment. Molecular targeting is one of the new therapeutic approaches to target cancer. Among a plethora of targetable signaling molecules, PDK1 is currently rising as a potential target for cancer therapy. Its aberrant expression in many malignancies is observed associated with glycolytic re-programming and chemo/radioresistance. **Methods**: Furthermore, to better understand the role of PDK1 in OSCC, we analyzed tissue samples from 62 patients with OSCC for PDK1 expression. Combining in silico and in vitro analysis approaches, we determined the important association between PDK1/CD47/LDHA expression in OSCC. Next, we analyzed the effect of PDK1 expression and its connection with OSCC orosphere generation and maintenance, as well as the effect of the combination of the PDK1 inhibitor BX795, cisplatin and radiotherapy in targeting it. **Results**: Immunohistochemical analysis revealed that higher PDK1 expression is associated with a poor prognosis in OSCC. The immunoprecipitation assay indicated PDK1/CD47 binding. PDK1 ligation significantly impaired OSCC orosphere formation and downregulated Sox2, Oct4, and CD133 expression. The combination of BX795 and cisplatin markedly reduced in OSCC cell’s epithelial-mesenchymal transition, implying its synergistic effect. p-PDK1, CD47, Akt, PFKP, PDK3 and LDHA protein expression were significantly reduced, with the strongest inhibition in the combination group. Chemo/radiotherapy together with abrogation of PDK1 inhibits the oncogenic (Akt/CD47) and glycolytic (LDHA/PFKP/PDK3) signaling and, enhanced or sensitizes OSCC to the anticancer drug effect through inducing apoptosis and DNA damage together with metabolic reprogramming. **Conclusions**: Therefore, the results from our current study may serve as a basis for developing new therapeutic strategies against chemo/radioresistant OSCC.

## 1. Introduction

Oral cancer is one of the most prevalent malignancies worldwide. Oral cancer-associated mortality continues to rise, with a predicted global mortality rate of 67.1%; this makes it one of the major causes of cancer-related deaths [1]. The most common histological subtype of oral cancer is oral squamous cell carcinoma (OSCC), which is highly aggressive, chemoresistant, and often allied with relapse and poor prognosis [1,2]. Currently, the treatment choice for OSCC is surgical resection, followed by radiotherapy and/or chemotherapy. However, even with an aggressive chemotherapeutic approach, median survival rates of patients with OSCC remain low, with an estimated 5-year survival of less than 50% [3]. Thus, a better therapeutic approach is urgently required.

Molecular targeting is a new therapeutic approach in cancer. Among a plethora of targetable signaling molecules, PDK1 is currently rising as a potential target for cancer therapy, its aberrant expression in malignancies is associated with chemoresistance [4,5]. The PDK1 protein is encoded by the *PDPK1* gene, a member of the AGC kinases, located at 16p13.3. In breast and prostate cancer, the amplification of this locus has been associated with poor prognosis [6,7]. PDK1 amplifies PTEN, PIK3CA, and ERB2 signaling output to Akt, thereby increasing malignant cells’ resistance to PI3K pathway inhibitors [8]. PDK1′s strong connotation with the Akt signaling pathway was highlighted as early as 1997, following the discovery of the Akt molecule. In the presence of PtdIns(3,4,5)*P*_3_ and PtdIns(4,5)*P*_2_, PDK1-induced activation of Akt increases [9]. PDK1 has also been implicated in cancer invasiveness as it plays a significant role in cell migration and invasion, especially in three-dimensional environments [10,11]. Taken together, these findings imply PDK1 can be strongly involved in modulating cancer stem cells (CSCs).

CSCs are the population of cancer cells, that is similar to stem cells, can regenerate and proliferate indefinitely. They are responsible for cancer metastasis and chemoresistance, making them prime elimination targets [12,13]. Recent evidence suggests that PDK1 plays a vital role in CSCs phenotype maintenance. Inhibition of the CDK1/PDK1/β-catenin signaling axis downregulates CSCs phenotype formation and enhances hepatocellular cancer sensitivity to sorafenib therapy [14]. PDK1 directly induces PLK1, and subsequently, MYC to drive oncogenic transformation and CSCs renewal [15]. Another study on gliomas indicated that temozolomide resistance results from PDK1- and CHK1-induced modulation of CSCs [16]. 

Importantly, the role of metabolic reprogramming has been widely accepted as a hallmark of cancer [17]. Accumulating evidence suggests CSCs utilizes metabolic programming to maintain unlimited self-renewal potential adaptation and changes in the tumor environment [18]. With metabolic re-programming, CSCs cells favor rapid energy production, anabolic process and therapy resistance [18]. In OSCC, RNA sequencing analysis from mouse tongue showed a significant change in the “Metabolic Process” through altering the glycolysis and lipid metabolism in cancer groups [19], with the induction in the process of glycolysis and lipid process with the changes in the expression of *LDHA*, *PDK3*, *SLC2A1*, *PFKP*, *ACC1* and *FASN* genes in oral cancer observed [19]. 

Furthermore, the radioresistance tumor phenotype that has been observed is due to a combination of numerous factors, such as enhanced DNA repair (*γH2AX* expression) [20], alteration of cell cycle progression, free radicals and reactive oxygen species (ROS) scavenging. These internal radioprotective mechanisms give a survival advantage to these radioresistant cells, ultimately leading to treatment failure [21]. A few studies have explored the association between Akt signaling and CD47 [22,23]. CD47, known as an integrin-associated signal transducer or protein (IAST or integrin-associated protein (IAP)), belongs to the immunoglobulin (Ig) superfamily that provides a “self” or “doesn’t eat me” signal to the immune cells [24], possibly enabling cancer cells to hide from immune cells and avoid elimination. Furthermore, from our previous study, we reported and validated that the ablation of expression of CD47 modulates the OSCC pluripotency capabilities i.e., CSCs generation, and attributes to OSCC radioresistance [25].

In the present study, we investigated and identified the effect of PDK1 expression role in oncogenic (PDK1/Akt/CD47) and glycolytic (LDHA/PFKP/PDK3) axis on stemness feature of OSCC-CSC modulation, and then evaluated the effect of novel PDK1 inhibitor BX795 on improving chemo- and radiotherapy via tuning the downstream *CD47*/*Akt*/*PDK3*/*LDHA* gene expression. Subsequently, we examined the modulation of PDK1 expression in OSCC and tested whether targeting PDK1 sensitizes OSCC towards chemoresistance/radiotherapy resistance through a combination of in silico and in vitro analysis.

## 2. Result

### 2.1. Aberrant PDK1 Expression in Human OSCC Positively Correlates with Disease Progression

Previous studies have identified that activation of CD47 could contribute to the activation of PI3K/PDK1/AKT/mTOR oncogenic signaling [26,27]. To further validate the important association of this pathway with PDK1 (encode *PDPK1*), we explored the potential role of targeting CD47 through PDK1, we first examined *CD47* and *PDPK1* gene profiles in OSCC using the GEO dataset (GSE2280 and GSE30784). Heatmap profiling indicated *CD47* and *PDPK1* mRNA expression were elevated simultaneously (Figure 1A,B). The association between *CD47* and *PDPK1* was found varying degrees of positive correlations in both GSE2280 (R^2^ = 0.3917, *p* < 0.01) and GSE30784 (R^2^ = 0.04565, *p* < 0.01) (Figure 1C,D). Indicated the regulation link between PDK1 (*PDPK1*) and *CD47*. A tissue array constructed using OSCC tissue samples from 62 patients was studied to determine the expression of the PDK1 protein using IHC staining. Table 1 shows the TNM classification and degree of differentiation of the human samples between PDK1 expression and clinicopathological variables of TSGH OSCC patients. The PDK1 staining intensity gradually increased from early stages (stage I–II) to late stages (stage III-IV) compared with normal tissue (Figure 1E). In addition, Patients with higher PDK1 expression have poor OS survival than the ones with low PDK1 (Figure 1F). These findings corroborated with clinicopathological univariate analysis, which indicated that American Joint Committee on Cancer (AJCC) stages (Wald Chi-Square test, *p* = 0.054), M stages (Wald Chi-Square test, *p* = 0.019), and PDK1 expression (Wald Chi-Square test, *p* = 0.004) was strongly associated with patient 5-years overall survival (Table 2). Consistently, multivariate analysis revealed that enhanced PDK1 expression was a potential predictor of patient survival (hazard ratio (HR) and 95% confidence interval (CI) = 0.278 (0.115–0.669), *p* = 0.004), along with AJCC stage (Wald Chi-Square test, *p* < 0.001) and M stages (Wald Chi-Square test, *p* = 0.014).

### 2.2. PDK1 Expression Associated with CD47 Modulation through Akt/Glycolysis Signaling Axis Activation

To understand the correlation between CD47 and PDPK1 in protein level, the CD47 expression in TSGH OSCC was analyzed by IHC and gradually increased from early stages (stage I–II) to late stages (stage III–IV) compared with normal tissue (Figure 2A). From the in silico analysis, we again observed a significant positive correlation (R^2^ = 0.09373, *p* < 0.05) of PDK1 and CD47 protein expression in clinical OSCC patients’ samples (Figure 2B). Next, protein-protein interaction analysis using STRING, demonstrated the PDK1 (*PDPK1*) has an average local clustering coefficient of 0.779, reflecting a significant and strong association of PDK1-CD47 (*p* = 0.0267) (Figure 2D). Interestingly, GEO analysis of the dataset (GSE30784) [28] of OSCC samples (tumor = 167 and normal = 45), using GEO2R online tool [29], the volcano plot showing differentially expressed genes (Appendix A) in the OSCC samples compared to normal (*p* < 0.05), the expression of the glycolytic gene (*LDHA*, *PDK3* and *PFKP*) were up-regulated in the OSCC samples. The Gene Set Enrichment Analysis (GSEA) indicated that the DEGs’ were enriched in a set of pathways, including glycolysis, fatty acid metabolism, mTOR signaling, inflammatory response, and TNFα/IL-6/JAK-STAT3-signalling (Appendix A) in OSCC samples. The protein-protein interaction analysis (STRING) showed AKT1 has an average local clustering coefficient of 0.745, reflecting a significant and strong association of PDK1-AKT1-LDHA (*p* = 5.44 × 10^−5^) (Appendix A). Further, to validate the aforementioned findings, we used two siRNA-PDK-1 clones: siPDK1-1 and siPDK1-2. we transfected SAS and TW2.6 cell lines with siPDK1-1 and -2, and analyze differential protein expression in the oncogenic (PI3K/mTOR/Akt) and glycolytic (LDHA/PFKP/PDK3)-signaling by Western blotting. PDK1 silencing markedly reduced the oncogenic signaling mediated by p-Akt, p-mTOR and p-PI3K, as well as glycolytic signaling mediated by LDHA, PFKP and PDK3 expression, compared with the control, which indicated that the PI3K/mTOR/Akt/LDHA/PFKP/PDK3- signaling pathway was inhibited through PDK1 inhibition (Figure 2D). Additionally, CD47 expression was also reduced when PDK1 expression was silenced. To further elucidate the molecular mechanism, we conducted immunoprecipitation of the SAS cell line against Akt with/without siPDK1 transfection and found that PDK1 was required to activate Akt on 473S and that Akt directly affected CD47 expression through PDK1 signalling (Figure 2E). Taken together, our findings demonstrate that PDK1 is strongly associated with CD47 expression by modulating the Akt/Glycolytic signaling axis.

### 2.3. PDK1 Silencing Attenuates CSC’s Phenotypes Maintenance in OSCC

Next, we hypothesized that PDK1 may play a crucial role in CSC’s modulation in OSCC as CD47 does. Herein, siPDK-1 and -2 transfected SAS and TW2.6 cells after successful PDK-1 knockdown, the transfected cells were grown in low-adhesion plates and incubated in a sphere enrichment medium. The siPDK-1 and -2 transfection significantly reduced SAS and TW2.6 primary orosphere formation (Figure 3A). Furthermore, we dissolved the primary spheres and regrew them in a new enrichment medium to generate a second generation of orosphere. Similar to the previous experiment, orosphere formation in cells with PDK1 ablation was significantly reduced compared with that in control (WT) cells (Figure 3B). To further understand these key mechanisms, we used the immunofluorescence assay to visualize CSC marker staining on OSCC orospheres; we found that siPDK1 elicited weaker Sox2 and Oct4 signals on both the SAS and TW2.6 orosphere (Figure 3C). Furthermore, the protein expression level of key CSC markers in these siPDK1 attenuated sphere cells was analyzed compared to WT control cells. siPDK1 significantly co-suppressed CD47, Sox2, Oct4, and CD133 expression compared with control in both the OSCC cells (Figure 3D). These findings suggest that PDK1 inhibition may negatively affect the expression of CSC phenotype through CD47 inhibition.

### 2.4. Pharmacologic Inhibition of PDK1 Acts Synergistically with Cisplatin in the Human OSCC Cell Line

As CSCs were reported to induce OSCC chemoresistance, herein, we investigated the anticancer effect of combining PDK1 inhibitor BX795 and the cisplatin, a conventional OSCC chemotherapeutic agent. OSCC cell lines were treated with various doses of BX795 (0.5–5 μM) and evaluated their viability using the sulforhodamine B (SRB) assay. BX795-IC_50_ values in SAS and TW2.6 cell lines were 29 and 37 μM, respectively. Next, we incubated SAS and TW2.6 cells simultaneously with BX795 and cisplatin for 48 h and evaluated their effect by using the SRB assay. OSCC cells exposed to BX795 combined with cisplatin exhibited a considerably lower proliferation rate (Figure 4A). The Chou–Talalay algorithm-based CompuSyn software was used for the isobologram-aided combinatorial analysis of BX795 with cisplatin; the result demonstrated that all of the combination points within the right-angled isobologram triangle had combination index scores of <1 for all corresponding dose combinations, indicating synergism between the two agents (Figure 4B). Moreover, the percentage of apoptotic cells stained with annexin V/7-AAAD in each group of treatment was detected by flow cytometry. In the combination treatment of BX795 and cisplatin, the SAS cells exhibited a significantly higher apoptosis rate of 30% and 27%, respectively than that with cisplatin (14% and 17%, respectively) or BX795 (19% and 18%, respectively) monotherapy (Figure 4C). To further explore whether the changes in the expression of pro-apoptotic (Bak and Bax), antiapoptotic (Bcl2x) and apoptotic (Cleaved-Caspase 3/9) factors in the combination treatment. Western-blot analysis assay was estimated and showed that Bak/Bax and Cleaved-Caspase3/9 protein expression was significantly induced while Bcl2 protein expression was reduced in combination treatment (Figure 4D). Interestingly, correlation analysis of PDK1 (*PDPK1*) gene expression to *BCL2* demonstrated a positive (R^2^ = 0.03588, *p* = 0.0017), whereas negative *BAX* (R^2^ = 0.5576, *p* < 0.001) correlation was detected in TCGA OSCC dataset (Figure 4E). These results highlighted the potential role of therapeutically targeting PDK1 to enhance the anticancer properties of cisplatin against OSCC.

### 2.5. BX795 and Cisplatin Combination Therapy Effectively Diminishes Human OSCC Cell Line Epithelial–Mesenchymal Transition Properties 

A synergistic relationship between BX795 and cisplatin was observed on OSCC cells, therefore, we used this combination dose to determine the effect on epithelial–mesenchymal transition (EMT) capabilities of OSCC cell lines. The oncogenic and tumorogenic potential that is the migratory and invasive potential of SAS and TW2.6 cells under the combined drug effect was evaluated. The migration and invasion ability of SAS and TW2.6 cells after treatment with cisplatin or BX795 or both were observed. We incubated SAS and TW2.6 cells in 24 wells until confluence and subsequently scratched the well surface using 200 µM pipette tips. After washing, we incubated the cells with fresh media containing cisplatin or BX795, or both. Although cisplatin or BX795 monotherapy significantly reduced wound healing of OSCC cells compared with control, their combination drastically reduced wound healing (Figure 5A). Furthermore, to evaluate how the treatment affects cell invasion, we incubated 2 × 10^4^ SAS and TW2.6 cells and analyzed the number of cells able to migrate through a gel-laden Transwell chamber after 24 h of treatment. Similar to the wound healing assay, the drug combination significantly inhibited the number of invading OSCC cells compared with control or monotherapy (Figure 5B). Additionally, to evaluate the effect of drug treatment on the self-renewal abilities of OSCC cells, we incubated 1 × 10^3^ SAS and TW2.6 cells in a 6-well plate treated with cisplatin, BX795, or both for 14 days to evaluate their colony-forming potential. The effect of cisplatin or BX795 monotherapy effectively inhibited OSCC cells’ colony-forming abilities but, the combination treatment was the most effective (Figure 5C). Western blotting analysis revealed that the drug combination synergistically modulates the expression of key EMT markers: N-cadherin, E-cadherin, Vimentin, Slug and Snail, together with the expression of β-catenin a key transcription factor of the EMT process (Figure 5D). Taken together, these findings imply that PDK1 can be a key therapeutic target in OSCC and may enhance the anti-EMT effect of cisplatin.

### 2.6. Inhibition of PDK1 Expression Enhances the Sensitivity of OSCC Cells toward Radiotherapy

To investigate the mechanisms underlying the effect of PDK1 inhibition on OSCC cells’ sensitization towards radiotherapy we examined the altered expression of PDK1 on the cell viability or proliferation of the SAS and TW2.6 OSCC cells, under the radiotherapy response (Figure 6A). After 48 h post-siRNA-PDK1 (siPDK1-1 and -2) transfection compared with control, these cells were exposed to the indicated value of radiation (0–15 Gy). The combination of PDK1 inhibition and radiotherapy significantly induces cell death in comparison to the radiation-only control group (Figure 6B). Subsequently, a clonogenic assay was used to identify the combination effect of radiotherapy and PDK1 inhibition on the self-renewal abilities of OSCC cells. The results revealed that radiation inhibited the clone counts in PDK1-depleted-SAS and TW2.6 OSCC cells compared to that of the control (Figure 6C), suggesting that PDK1-deficient OSCC cells were sensitized toward radioresistance. Furthermore, the mechanism through which ablation of PDK1 inhibits OSCC cell growth, we used flow cytometry to assess whether PDK1 affects the apoptosis and cell cycle progression. Flow cytometry analysis was used to detect the apoptosis of PDK1 depleted SAS and TW2.6 OSCC cells with or without radiation (0 Gy-5 Gy) after Annexin V/7-AAD staining. Irrespective of 0 Gy or 5 Gy irradiation, PDK1 silencing induces the rate of OSCC cells apoptosis (Figure 6D), and the number of apoptotic cells was notably more in the PDK1 knockdown cells compared to the control group. In comparison with the control group, 5 Gy irradiation in the siPDK1 groups causes a slightly larger accumulation of cells in the G2/M phase, while 0 Gy irradiation in the siPDK1 group showed a remarkably reduced proportion of G2//M phase (Figure 6E). These results demonstrated that PDK1 silencing can induce cell cycle arrest under the irradiation effect. Interestingly, we also analyzed the influence of PDK1 on DNA repair and damage, the DNA damage marker γH2AX an early cellular response marker to the induction of DNA double-strand breaks expression was evaluated by western blot analysis. Western blotting analysis showed that depletion of PDK1 promotes DNA damage after the irradiation. The protein level of γH2AX, CD47, p-PDK1, P-PI3K, p-Akt, p-mTOR, LDHA, PFKP, and PDK3 were reduced under the 5 Gy irradiation for 24 h (Figure 6F). Importantly, the effect of PDK1 inhibition combined with 5 Gy irradiation effectively reduces the tumorsphere (orosphere) generating ability of OSCC-CSC’s cells (Figure 6G). Overall, these results showed that PDK1 inhibition sensitizes OSCC cells towards radiotherapy via reducing the expression of DNA repair enzymes, confirming that inhibition of the DNA repair process induces the radiosensitivity of OSCC cells.

## 3. Discussion

At present, chemo/radiotherapy is one of the main treatment modalities for higher OSCC patients [30]. However, during the treatment time course often OSCC patients ultimately develop chemo/radioresistance [30]. Therefore, it is particularly vital to understand and explore the underlying mechanism of radioresistance in OSCC patients. 

Interestingly, we reported from our previous work that ablating the expression of CD47 modulates the OSCC pluripotency capabilities that is CSCs generation and attributes to OSCC radioresistance [25]. Through the in silico analysis of OSCC datasets, we find a potential association between PDK1 (*PDPK1*) and *CD47*, together with the expression of Akt1 and glycolytic *LDHA* gene. Therefore, in this study, we demonstrated PDK1 may act as a prognostic biomarker in OSCC tumors. Using the immunoprecipitation approach, we further confirmed and validated that CD47 expression was affected by PDK1 silencing through the Akt signaling axis. Furthermore, PDK1 silencing inhibits OSCC-CSCs maintenance and self-replication abilities by impairing glucose uptake. The combination of BX795, a PDK1 inhibitor, with cisplatin and radiotherapy markedly reduced OSCC cell migration and invasion capabilities, and induced apoptosis through altering γH2AX expression. The combination was shown to act synergistically, as quantified using the Chou–Talalay algorithm, with combination index scores of <1. In vitro analysis demonstrated, BX795 treatment effectively enhanced cisplatin/radiotherapy cytotoxicity through enhancing DNA damage and altering the oncogenic and glycolytic signaling via modulating the expression of PDK1/Akt/CD47/γH2AX and LDHA/PFKP/PDK3 signaling in OSCC cells.

The PI3K/Akt oncogenic signaling pathway is one of the most altered pathways in cancers [31], producing typical aberrant behaviors, such as aggressive invasion, increased proliferation, and resistance to apoptosis and therapy [31]. PDK1 or *PDPK1* phosphorylates Akt and many other AGC kinases such as p70 ribosomal protein S6 kinase (p70S6K), serum/glucocorticoid regulated kinase (SGK), p90 ribosomal protein S6 kinase (p90RSK), and members of the protein kinase C family [32]. PDK1 is also considered constitutively active in cancers because its activation site (serine 241) is activated by PDK1 itself and not by other kinases [33]. Uncontrolled cell proliferation, apoptosis evasion, invasion, metastasis, and abnormal angiogenesis are some of the pathological phenotypes associated with PDK1 activation [34]. Higher PDK1 expression was shown to confer a worse prognosis in various malignancies. Castration-resistant prostate cancers have a higher amplification of the locus containing the PDK1 gene than 25 primary tumors [35]. PDK1 is significantly highly expressed in esophageal squamous cell carcinoma compared with adjacent noncancerous tissues and associated with shorter OS [36]. Metastatic melanoma is associated with increased PDK1 expression [37]. These data are in line with our findings that higher PDK1 expression correlated with poor overall survival in OSCC patients.

The presence of CSCs is problematic due to their rapid differentiation (and thus the ability to form tumors), higher rate of cell migration and invasion, metastasis, and resistance to chemo-/radiotherapy [38]. PDK1 was implicated in the modulation of CSCs and glycolytic genes, especially CDK1-PDK1-PI3K/Akt signaling axis was reported to regulate pluripotency in stem cells themselves, making PDK1′s key role in CSCs modulation in OSCC [39]. PDK1 and AurA kinase inhibitors, MP7 and alisertib, respectively, acted synergistically in targeting glioma stem cells, inducing their differentiation and apoptosis [40]. In HCC, targeting the CDK1/PDK1/β-catenin pathway attenuated the pluripotency proteins such as Oct4, Sox2 and Nanog, and improved the efficacy of sorafenib against HCC cells [14]. The role of metabolic reprogramming is widely accepted as a hallmark of cancer [17]. CSCs hijack and reprogram the metabolic process to maintain its unlimited self-renewal potential and changes the tumor environment [18] to favor rapid energy production, anabolic process and therapy resistance [18]. The study demonstrated the expression of the glycolytic enzyme lactate dehydrogenase A (LDHA) regulates the expression of PI3K-Akt signaling results in the re-programing of bioenergetic mechanisms in breast and liver cancer progression [41,42].

The present study also demonstrated the suppression of OSCC orosphere generation after the PDK1 siRNA silencing. Because CSC’s are strongly associated with EMT, we further explored whether PDK1 affects EMT in OSCC cells. PDK1 suppression (using a novel PDK1 inhibitor BX795) significantly reduced the migratory and invasive properties of OSCC cells, and a combination of cisplatin/radiotherapy augmented this effect. Cancers treated with cisplatin/radiotherapy might eventually develop resistance to them and relapse; these cancers tend to be more aggressive and show a higher EMT process [43]. Controlling EMT with BX795 seemed to have been effective as CSCs maintenance was significantly inhibited, as demonstrated in our previous experiments. PDK1 silencing/inhibition results in the OSCC cells in the cell cycle (G2/M) arrest and induced apoptosis under the irradiation effect. Interestingly, the influence of PDK1 on DNA repair and damage was also seen, a reduction in the expression of the DNA damage marker γH2AX an early cellular response marker to the induction of DNA double-strand breaks expression was observed. In GBM and gall bladder cancer, the PDK1/Jun signaling pathway was shown to promote cell proliferation and the EMT process [44,45]. Recently, a flavonoid derivative, cyanidin, was reported to inhibit oxaliplatin-induced EMT by targeting the PDK1-PI3K/Akt pathway [46]. Notably, the PI3K/Akt/mTOR signaling pathway also plays a critical role in an immunosuppressive microenvironment, such as the expression of the immune checkpoint ligand PD-L1 [47]. However, few studies have discussed the relationship between Akt signaling and CD47.

CD47, formerly known as IAP, is expressed ubiquitously on the cell membrane [48]. It is an Ig V-like domain that interacts with various proteins and participates in various biologic processes such as leukocytes motility, platelet activation, and regulation of apoptotic cell clearance [48]. CD47 is highly expressed in malignancies and is considered to be an adverse clinical prognostic factor [49]. It mainly serves as an apoptosis evasion signal in tumors. Its interaction with SIRPa, a molecule expressed on macrophages, sends a “don’t eat me” signal that allows tumor cells to escape phagocytosis [50]. CD47 was also implicated in other tumor cell defense mechanisms. Ligation of CD47 induces type III apoptosis in various malignancies, suggesting a direct antitumor effect with CD47 inhibition [51,52]. CD47 is associated with drug resistance in tumor-initiating cells. Lee et al. found that blocking cathepsin through CD47 suppression resulted in a marked reduction in tumor-initiating cells maintenance and tumor initiation in hepatocellular cancer [53]. In addition, the miR-708/CD47 signaling pathway was found to play a crucial role in breast cancer stem cell-like phenotypes and drug resistance [54].

Recently, our group published a paper elucidating the role of CD47 in OSCC-CSCs modulation. We discovered that CD47 knockdown negatively influenced OSCC cell viability and orosphere formation. Additionally, CD47 silencing reduced EMT, migration, and clonogenicity of OSCC cells [25]. Following this study, we speculated whether CD47 can be affected by other molecules. Through STRING, protein–protein interaction analysis implied an unspecified association between CD47 and Akt, which was confirmed through the immunoprecipitation assay: downregulation of Akt through PDK1 silencing negatively affected CD47/LDHA expression. This finding confirmed Akt/CD47 binding and indicated the PDK1/Akt/CD47/LDHA signalling axis pathway as a viable therapeutic target in OSCC. Overall, these results showed that PDK1 inhibition sensitizes OSCC cells towards radiotherapy via reducing the expression of DNA repair and glycolytic enzymes, confirming that inhibition of the DNA repair and glucose uptake process induces the radiosensitivity of OSCC cells.

However, in astrocytoma, CD47 was shown to induce proliferation through Akt phosphorylation [55]. A similar event was also observed in glioblastoma, where cells overexpressing CD47 displayed increased Akt activation and invasion ability. Conversely, suppressing Akt signaling impaired GBM cells reversed this effect [23]. Thus, more studies are warranted to elucidate the relationship between these molecules.

## 4. Materials and Methods

### 4.1. Patient Samples

The prognostic relevance of PDK1 expression in patients with OSCC (*n* = 62) recruited from the National Defense Medical Center between October 2000 to March 2013. All participants provided written informed consent, and this study was reviewed and approved by the institutional review board (TSGHIRB 2–102-05–125). Patient distribution based on AJCC staging, 7th edition, aged between 29 and 72 years (median, 50 years). Pretreatment evaluations included a detailed clinical history, physical examination, barium swallow X-ray, upper gastrointestinal tract endoscopy, and computed tomography scans of the thorax and abdomen. All patients were treated according to the standard treatment of TSGH and NCCN guidelines. 

### 4.2. Immunohistochemical Staining

For the PDK1 immunohistochemical staining analysis of OSCC (*n* = 62) and nontumor oral tissues (*n* = 7), formalin-fixed paraffin-embedded tissue sections were blocked using 1% bovine serum albumin (BSA) and were then incubated with PDK1 antibody (Santa Cruz Biotechnology, Santa Cruz, CA, USA) at 4 °C overnight. Tissue staining was scored by two independent pathologists. For better determining the correlation of PDK1 and OSCC prognosis, the tissue samples were grouped by their disease stages (I, II, III, and IV). Tissues were stained with antibodies against PDK1 (1:100, E-3, sc-17765, Santa Cruz Biotechnology, Santa Cruz, CA, USA) at 4 °C overnight. Horseradish peroxidase (HRP) and diaminobenzidine (DAB) staining using the mouse and rabbit–specific HRP/DAB (ABC) detection kit (ab64264, Abcam, Cambridge, MA, USA) as well as hematoxylin counterstaining were performed according to the standard immunohistochemistry protocol, followed by imaging and evaluation of PDK1 expression. Percentage of positively stained tumor cells was graded as 0 (<10%), 1 (10–25%), 2 (25–50%), 3 (50–75%), and 4 (>75%), and staining intensity was graded as 0 (no staining), 1 (weak), 2 (moderate), and 3 (strong). The quick score (Q score) is defined as the percentage of staining cells (%) multiplied by a score of intensity. The membrane and cytoplasm scores were calculated in each. The total Q score was given as membrane score plus cytoplasm score. All enrolled patients provided written informed consent for their tissues to be used for scientific research. The study was approved by the Joint Institutional Review Board of TSGH Hospital (approval number: TSGHIRB 2–102-05–125) and was compliant with the Helsinki Declaration.

### 4.3. Cell Lines and Culture

The human OSCC cell lines SAS and TW2.6 were kindly provided by Prof. Hsiao Michael (Genomic Research Center, Academia Sinica). Both cell lines were cultured in Dulbecco’s modified Eagle’s medium (DMEM, Invitrogen Life Technologies), with 10% fetal bovine serum (FBS) supplementation and 1% penicillin/streptomycin (Invitrogen Life Technologies). Cells were incubated at 37 °C in a 5% humidified CO_2_ incubator. The OSCC cells were passaged at 70–90% confluence, and the medium was changed every 72 h. The cells were used at passage 5–20 for the experiments and were not grown beyond 20 passage.

### 4.4. siRNA Transfection of OSCC Cell Lines

SAS and TW2.6 cells were transfected with shRNA specifically targeting *PDPK1* or control/scramble siRNA purchased from Santa Cruz (Santa Cruz Biotechnology, Santa Cruz, CA, USA). The OSCC cells were transfected with siRNA following the manufacturer’s instructions. siRNA-PDK1 transfected clones were then expanded for future use.

### 4.5. Orosphere Formation and Self-Renewal Assay

Next, 5 × 10^4^ cells/well OSCC cells were seeded in 6-well nonadherent plates (Corning Inc., Corning, NY, USA) in serum-free DMEM/F12 medium (11330057; Thermo Fisher Scientific, Bartlesville, OK, USA) supplemented with basic fibroblast growth factor (bFGF) (20 ng/mL, Invitrogen Life Technologies, Carlsbad, CA, USA), B27 supplement (Invitrogen Life Technologies), 5 μg/mL insulin (91077C; Sigma Aldrich, St. Louis, MO, USA), and epidermal growth factor (EGF) (20 ng/mL, Millipore, Bedford, MA, USA). After 12–15 days, orospheres were counted using an inverted phase-contrast microscope. Next, primary orospheres consisting of ≥50 μm spheres were counted, and images were taken with the microscope. For secondary orosphere generation, primary orospheres were dissociated through trypsinization and then pipetted to obtain a single-cell suspension using a 22-G needle (Thermo Fisher Scientific, Bartlesville, OK, USA). After 12–15 days of culture, secondary orospheres consisting of ≥50 µm were counted, and images were taken under the microscope.

### 4.6. Cell Cycle and Apoptosis Analysis

After all the treatment cells were irradiated with 0 Gy-5 Gy with 6 MV X-rays for 48 h, 6 × 10^5^ cells were collected from each group and washed with PBS buffer. Each group of cells were fixed with precooled 75% of 1 mL ethanol. Then, 0.5 mL PI/RNA staining buffer (BD, Franklin Lakes, NJ, USA) was added for 20 min. All cells were then filtered with a nylon filter and detected by a flow cytometer. For apoptotic analysis, after the irradiation and PBS wash, 6 × 10^5^ cells suspension were added 5 µL Annexin V/7-AAD (BD, NJ, USA) for 20 min. Thereafter, these stained cells were analyzed and detected under a flow cytometer. 

### 4.7. Sulforhodamine B Cytotoxicity Assay

Drug cytotoxicity and cell viability were assessed using the SRB assay, as previously described [56]. First, aliquots of 3 × 10^3^ OSCC cells were seeded in 96-well plates, with each well containing the supplemented medium, and incubated in humidified 5% CO_2_ in air at 37 °C for 24 h. Next, OSCC cells were treated with various concentrations of BX795 or cisplatin, and untreated cells served as controls. Quantification was performed twice in triplicate experiments. Optical density was measured at a wavelength of 495 nm by using a SpectraMax microplate reader (Molecular Devices, Kim Forest Enterprises Co., Ltd., Taipei, Taiwan).

### 4.8. Western Blot Analysis

For western blotting, cells were lysed using RIPA buffer (Cell Signaling Technology) with a cocktail of protease inhibitors (Sigma Aldrich). Next, 10 μg of protein samples were separated using 10% SDS-PAGE electrophoresis and transferred to polyvinylidene fluoride (PVDF) membranes using the Bio-Rad Mini-Protein electro-transfer system (Bio-Rad Laboratories, Hercules, CA, USA). Next, nonspecific binding was blocked by incubating the membranes in 5% BSA in Tris-buffered saline with 0.1% Tween 20 (TBST) for 1 h at room temperature, and then the membranes were incubated overnight at 4 °C with the antibodies (Appendix A). The membranes were later incubated with the corresponding HRP-conjugated anti-rabbit IgG or anti-mouse IgG secondary antibodies (Cell Signaling Technology) for 1 h at room temperature and washed with PBS four times. Band detection was subsequently performed using enhanced chemiluminescence, western blotting reagents, and a BioSpectrum Imaging System (UVP; Upland, CA, USA). ImageJ v. 1.46 (https://imagej.nih.gov/ij/, accessed on 19 October 2021) was used to quantify the bands. Target protein expression was normalized to that of GAPDH, and assays were repeated four times in triplicates.

### 4.9. Wound Healing Migration Assay

OSCC cells were seeded in 24-well plates and cultured to 100% confluence. Next, a 1-mm wound was made along the median axis of the well using a 200-µL pipette tip. Cell migration into the wound area was then observed at 0 and 24 h under the microscope. The assay was performed three times in triplicates.

### 4.10. Matrigel Invasion and Migration Assay

Cell invasion assays were performed in Boyden chambers (pore size = 8 μm) with the upper side of the filter covered with 0.2% Matrigel diluted in DMEM. Then, 1 × 10^4^ OSCC cells in serum-free culture medium treated with cisplatin, BX795, or both were plated in the upper chambers, whereas the lower chambers contained a complete culture medium with 10% FBS as a chemoattractant. After overnight incubation, the un-invaded OSCC cells on the upper side of the filter were carefully removed with a cotton bud, whereas the invaded cells on the lower side of the membrane were washed, fixed in 95% ethanol, and stained with 10% Giemsa dye. Four randomly selected fields of each membrane were used to count the invaded cells and averaged to obtain a representative number of the invaded cells.

### 4.11. Immunofluorescence Staining

For the immunofluorescence analysis, the OSCC cells were plated in 6-well chamber slides (Nunc, Thermo Fisher Scientific, Waltham, MA, USA) for 24 h, fixed in 2% paraformaldehyde at room temperature for 10 min, permeabilized with 0.1% Triton X-100 in 0.01 M PBS (pH 7.4) containing 0.2% BSA, air-dried, and rehydrated in PBS. The cells were then incubated with antibodies against c-Myc and Oct4 diluted 1:200 in PBS containing 3% normal goat serum at room temperature for 2 h, followed by incubation with anti-rabbit IgG fluorescein isothiocyanate-conjugated secondary antibody (Jackson ImmunoResearch Lab, West Grove, PA, USA). After resting at room temperature for 1 h, cells were washed in PBS and mounted using the Vectashield mounting medium while counterstaining with 4′,6-diamidino-2-phenylindole (DAPI, D3571, Molecular Probes, Life Technologies, Carlsbad, CA, USA). Images were captured using a Zeiss Axiophot fluorescence microscope (Carl Zeiss, Hsinchu City, Taiwan).

### 4.12. Statistical Analysis

Each experiment was performed at least three times in triplicates. All statistical analyses were conducted using IBM SPSS Statistics for Windows, Version 25.0 (IBM, Armonk, NY, USA). All data are presented as means ± standard deviation (SD). Comparison between two groups was estimated using the two-sided Student’s *t*-test, and one-way analysis of variance was used for comparison between three or more groups. The association between the differential expression of PDK1 and OS in patients with OSCC was determined using univariate Cox proportional regression of covariates, including age, sex, AJCC stage, pathological grade, local recurrence, and lymph node involvement. Variables for which *p* < 0.05 were identified as significantly associated with prognosis, and Cox multivariate analysis was subsequently performed for these variables. HR and 95% CI for multivariate analyses were computed using the Cox proportional hazards regression. *p* < 0.05 was considered statistically significant.

## 5. Conclusions

In conclusion, as described in the graphical abstract (Figure 7, below), PDK1 suppression inhibited Akt/CD47/LDHA signaling cascade and reduced OSCC cells EMT and orosphere maintenance abilities. This inhibition ultimately resulted in targeting glycolytic pathways which resulted in the enhanced chemo and radiosensitivity of OSCC cells. Overall, our work laid the basis for further exploration of the clinical feasibility of targeting PDK1 as a therapeutic molecular target in oral cancer.

## Figures and Tables

**Figure 1 ijms-22-11492-f001:**
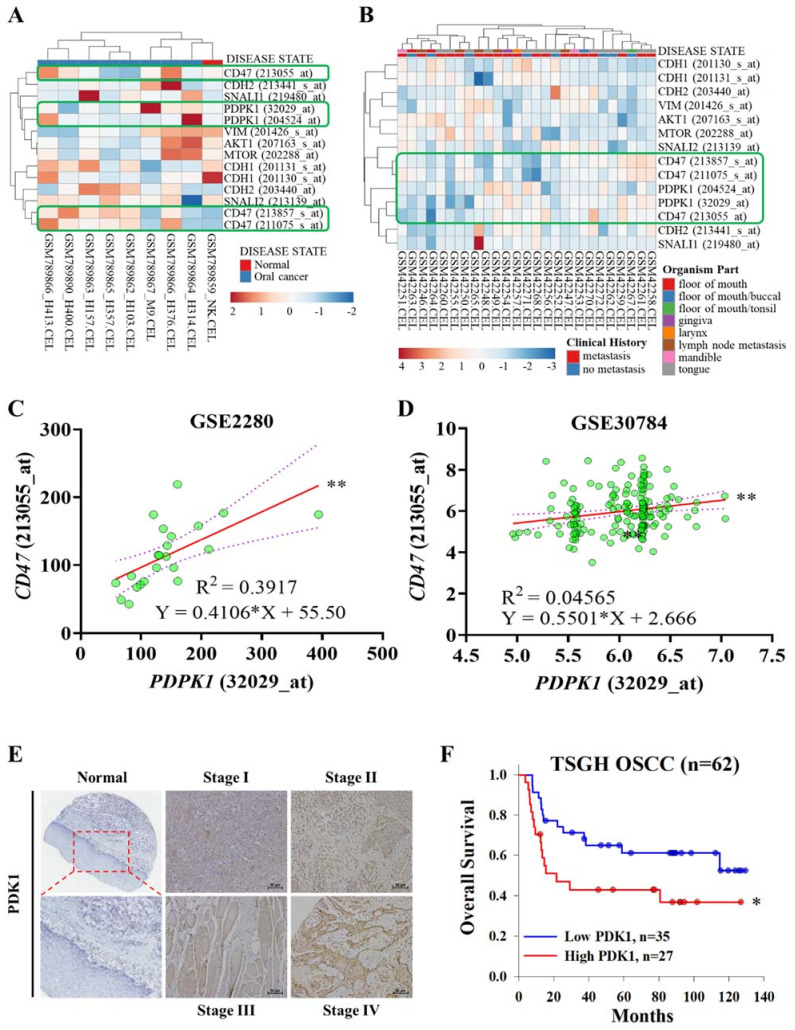
PDK1 confers a poor prognosis in OSCC. (**A**,**B**) Heatmap of GSE2280 and GSE30784 gene expression profiling. (**C**) Graphical representation of *PDPK1* and *CD47* mRNA coexpression analysis in GSE2280 with R^2^ = 0.4237, *p* < 0.001 and (**D**) in GSE30784 with R^2^ = 0.04565, *p* < 0.01. (**E**) Immunohistochemical staining of PDK1 in TSGH OSCC tissue categorized based on stages. (**F**) PDK1 expression for predicting patients with overall survival in TSGH OSCC (*n* = 62). The 5-year OS was 61.0% and 42.9% in BTK-low (*n* = 35) and BTK-high (*n* = 27) OSCC patients, respectively (*p* < 0.05). Q-score < 140 indicated BTK-low and Q-score ≥ 140 indicated BTK-high. * *p* < 0.05 and ** *p* < 0.001; scale bar 50 μm.

**Figure 2 ijms-22-11492-f002:**
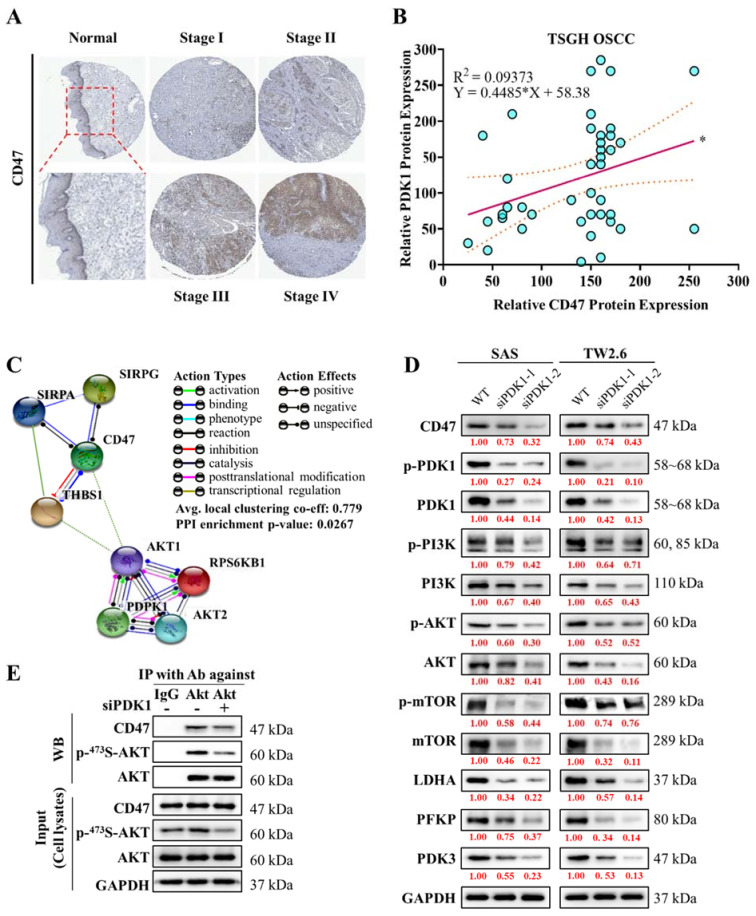
PDK1 is associated with increased *CD47* expression through the downstream AKT and Glycolysis signaling pathway. (**A**) Immunohistochemical staining of *CD47* in TSGH OSCC tissue categorized based on stages. (**B**) Graphical representation of PDK1 and *CD47* protein coexpression analysis in TSGH OSCC with R^2^ = 0.09373, *p* < 0.05. (**C**) STRING database analysis of the relationship between PDK1 (*PDPK1*), *Akt*, and *CD47* genes. Network nodes represent proteins encoded by the genes. (**D**) Western blotting showing differential protein expression of p-PDK1, PDK1, PI3K, p-PI3K, AKT, p-AKT, MTOR, p-MTOR, LDHA, PFKP, PDK3 and *CD47* between siControl and PDK1 silencing through siPDK1-1 and siPDK1-2 transfection. (**E**) Co-immunoprecipitation analysis reveals the association of PDK1 and *CD47* through Akt in SAS cells. SAS cells were transfected with siPDK1 or siControl and then equal amounts of cell lysates were immunoblotted (WB) with antibodies against *CD47*, p-Ser473-Akt, or Akt (Input; lower panel), or were immunoprecipitated (IP) with anti-Akt antibody and protein A/G agarose followed by immunoblotting with antibodies against CD47, p-Ser473-Akt, or Akt (upper panel). Big data analysis revealed the potential prognostic role of assessing. All data are representative of the experiment conducted four times in triplicate and are expressed as mean ± SEM. * *p* < 0.05.

**Figure 3 ijms-22-11492-f003:**
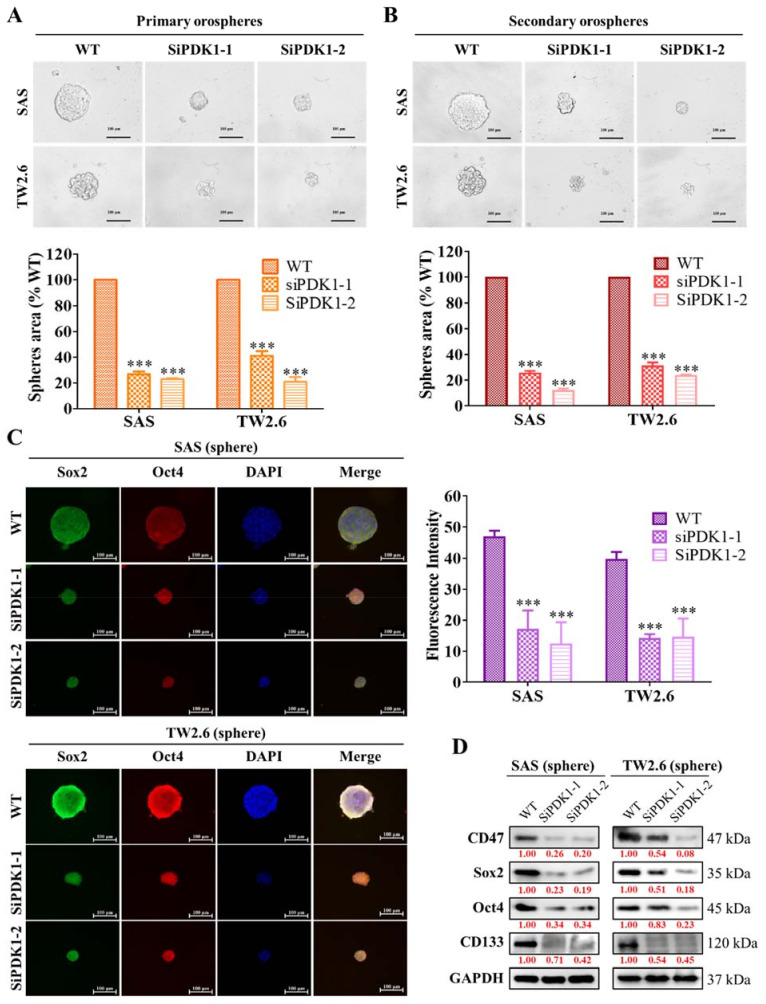
PDK1 modulates CSC-like phenotypes in OSCC cells. (**A**) Primary and (**B**) secondary orosphere generation of SAS and TW2.6 cell lines were reduced after transfection with siPDK1-1 and siPDK1-2. (**C**) Immunofluorescent staining showing the effect of PDK1 silencing on the expression of Sox2 and Oct4 proteins in spheres formed by SAS and TW2.6 cells. (**D**) Effect of PDK1 knockdown on the expression level of *CD47*, Sox2, Oct4, and CD133 proteins in SAS and TW2.6 sphere. GAPDH served as a loading control. All assays are representative of experiments performed four times in triplicates. WT, wild type; Sp, orosphere; blue stain = DAPI, nuclear staining. All data are representative of the experiment conducted four times in triplicate and are expressed as mean ± SEM. *** *p* < 0.001; scale bar 100 μm.

**Figure 4 ijms-22-11492-f004:**
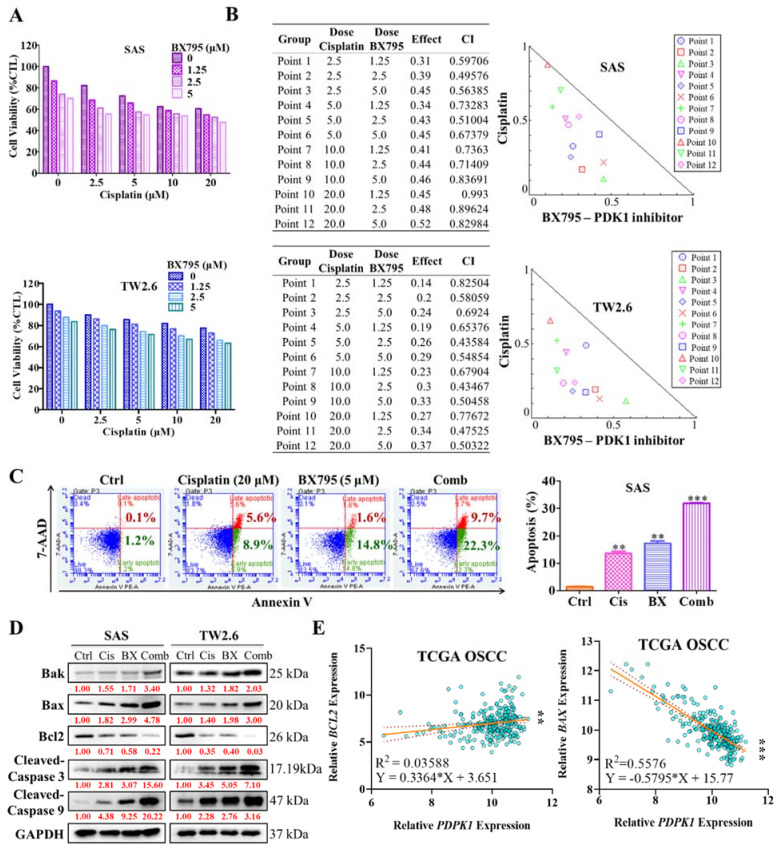
Synergistic anticancer effect of BX795 and cisplatin against OSCC cells. (**A**) Dose-response graphical charts of SAS and TW2.6 OSCC cells to BX795 and Cisplatin combination therapy show cell viability. (**B**) Combination effects were quantified using Compusyn software following the Chou–Talalay algorithm (Left). Representative isobologram figures of the combined potential of cisplatin and BX795 quantified through SRB assay (Right). Combinational effects are presented as the combination index (CI), where CI < 1 indicates synergism (inner triangle), CI = 1 (on the hypotenuse) indicates an additive effect, and CI > 1 (outer triangle) indicates antagonism. Data represent as mean ± SEM of three individual experiments, ** *p* < 0.01, *** *p* < 0.001. (**C**) SAS Cells were treated with cisplatin (20 µM), BX795 (5 µM), or their combination stained with annexin V/7-AAD before flow cytometry analysis. (**D**) The expression level of pro-apoptotic (Bak and Bax), antiapoptotic (Bcl2x) and apoptotic (cleaved caspase3/9) marker in OSCC cells as shown by Western blot analyses. (**E**) Correlation analysis of PDK1 (*PDPK1*) with pro-apoptotic markers and anti-apoptotic markers, *BCL2* and *BAX*, respectively.

**Figure 5 ijms-22-11492-f005:**
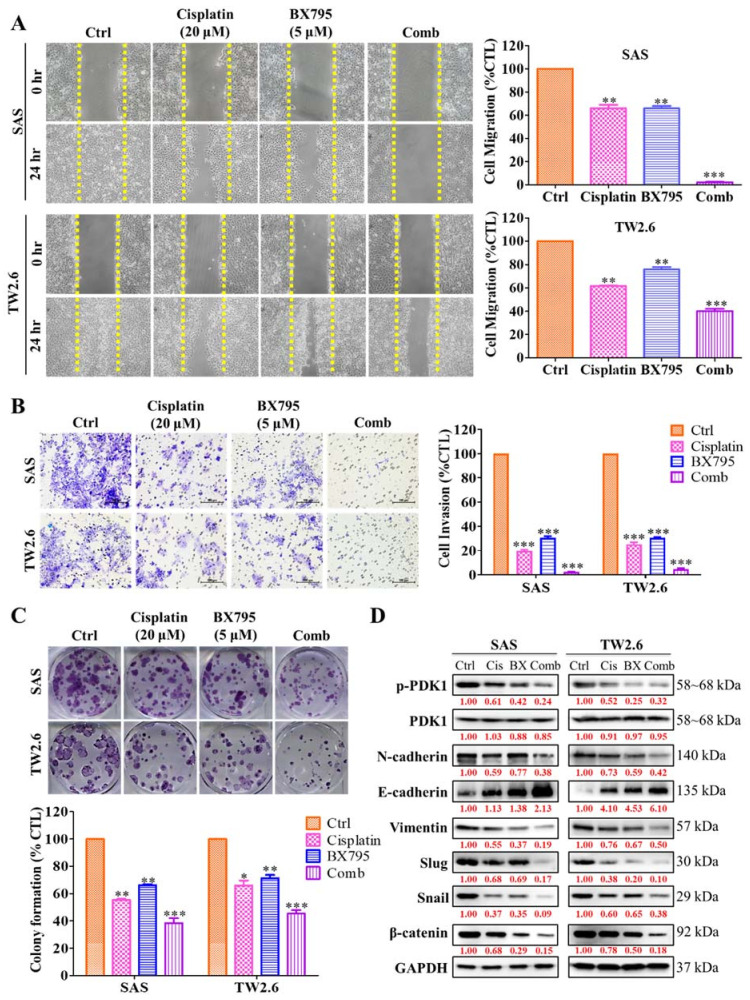
The combination of cisplatin and BX795 reduces OSCC cell migration, invasion, and colony formation. (**A**) A representative image of the scratch-wound migration assay shows the inhibitory effect of cisplatin, BX795, or their combination on the motility of SAS and TW2.6 cells at 24 h. (**B**) Representative images of SAS and TW2.6 cells invasion capabilities after treatment with cisplatin, BX795, or their combination. (**C**) Representative images of colonies formed by SAS and TW2.6 cells after treatment with cisplatin, BX795, or their combination in the culture plate were observed using crystal violet solution staining. (**D**) The inhibitory effect of cisplatin, BX795, or their combination on the expression of PDK1, p-PDK1, Vimentin, Slug, Snail, N-cadherin, E-cadherin and β-catenin in SAS and TW2.6 cells, as demonstrated by western blot analyses. GAPDH served as a loading control. All data are representative of the experiment conducted four times in triplicate and are expressed as mean ± SD * *p* < 0.05, ** *p* < 0.01, *** *p* < 0.001; scale bar 100 μm.

**Figure 6 ijms-22-11492-f006:**
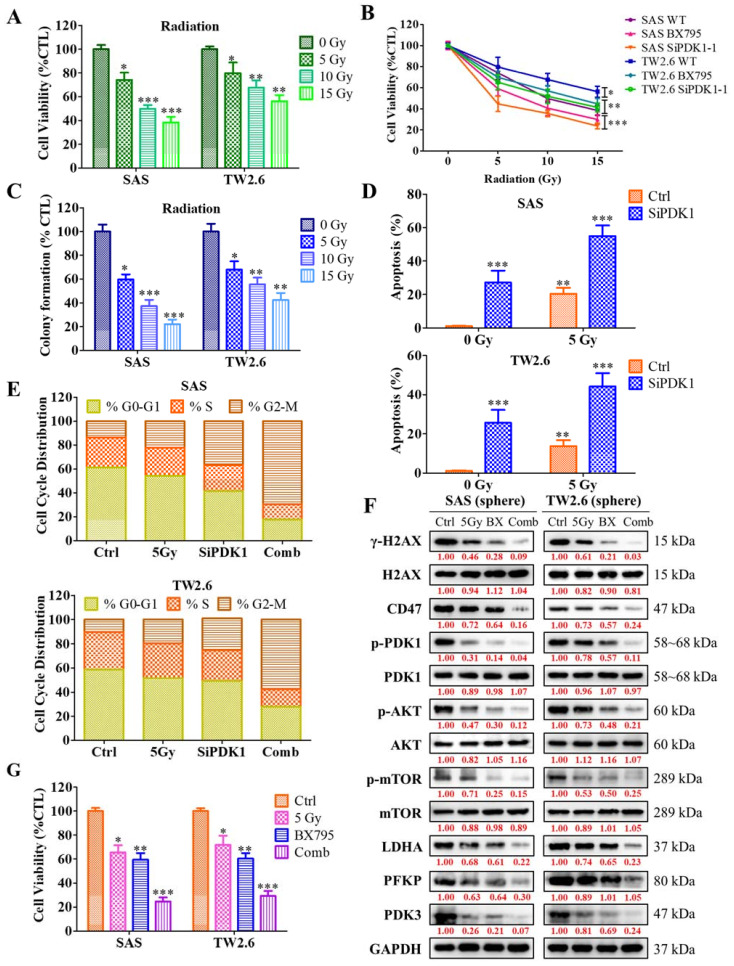
Suppression of PDK1 expression induces the sensitivity of OSCC cells toward radiotherapy. (**A**) Representative bar plot showing the inhibitory effect of radiotherapy (0 Gy-15 Gy) on the cell viability of SAS and TW2.6 OSCC cells. (**B**) The survival curve represented the radiosensitivity of PDK1 depleted (siPDK1-1, -2) and control (Wild type) SAS and TW2.6 OSCC cells. (**C**) The colony-forming assay was conducted for PDK1 knockdown and control OSCC cells with indicated IR dose. (**D**) Flow cytometry analysis was used to detect the apoptosis of PDK1 depleted SAS and TW2.6 OSCC cells with or without radiation (0 Gy-5 Gy). (**E**) The cell cycle analysis (G2/M) was evaluated by flow cytometry with or without 5 Gy radiation on PDK1 knockdown SAS and TW2.6 OSCC cells. (**F**) The inhibitory effect of 5 Gy, BX (BX795) or combination on the expression of the protein level expression of γH2AX, CD47, PDK1, Akt, mTOR, LDHA, PFKP, PDK3 in SAS and TW2.6 cells, as demonstrated by western blot analyses. GAPDH served as the internal control. (**G**) Effect of aforementioned treatment on the Orosphere generation ability of SAS and TW2.6 cell lines. * *p* < 0.05, ** *p* < 0.01, *** *p* < 0.001.

**Figure 7 ijms-22-11492-f007:**
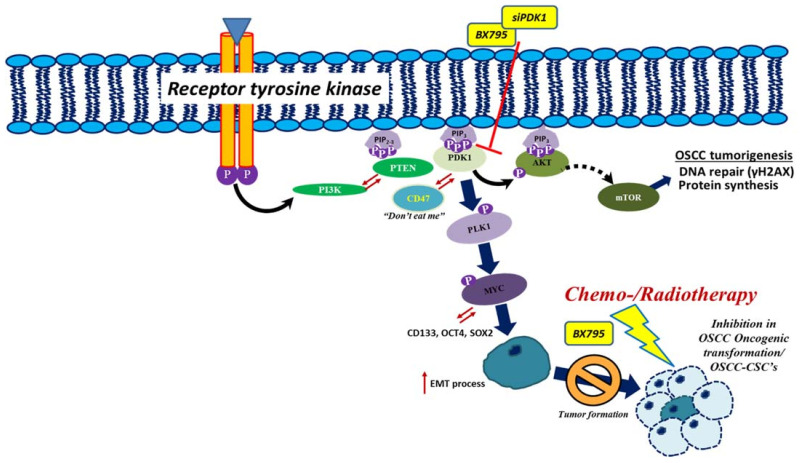
Graphical abstract.

**Table 1 ijms-22-11492-t001:** TNM classification, PDK1 expression and clinicopathological variables of TSGH OSCC patients.

Clinicopathological Variables	No. (*n* = 62)	PDK1		x^2^	*p*-Value
		High Expression	Low Expression		
**Median Age**				0.104	0.747
≤50	33	15 (45)	18 (55)		
>50	29	11 (41)	17 (59)		
**Gender**				0.112	0.737
Male	56	24 (43)	32 (57)		
Female	6	3 (50)	3 (50)		
**AJCC stage**				8.308	0.040
I	8	2 (25)	6 (75)		
II	16	3 (19)	13 (81)		
III	14	8 (57)	6 (43)		
IV	24	14 (58)	10 (42)		
**T stage**				2.410	0.492
1	12	3 (25)	9 (75)		
2	28	13 (46)	15 (54)		
3	5	3 (60)	2 (40)		
4	17	8 (47)	9 (53)		
**LN metastasis**				0.230	0.632
No	32	13 (41)	19 (59)		
Yes	30	14 (47)	16 (53)		
**M**, **Metastasis**				5.018	0.025
No	53	20 (38)	33 (62)		
Yes	9	7 (78)	2 (22)		
**Primary site**				4.525	0.718
Buccal	26	13 (50)	13 (50)		
Palata	1	0 (0)	1 (100)		
Tongue	27	10 (37)	17 (63)		
Gingiva	6	2 (33)	4 (67)		
Mouth Floor	0	0 (0)	0 (0)		
Maxilla	0	0 (0)	0 (0)		
Lip	1	1 (100)	0 (0)		
Tonsil	1	1 (100)	0 (0)		

**Table 2 ijms-22-11492-t002:** American Joint Committee on Cancer (AJCC) OSCC staging criteria and patient 5-years overall survival.

Univariate								Multivariate						
Covariate	Coefficient	SE	Wald	*p*-Value	HR	95%CI		Coefficient	SE	Wald	*p*-Value	HR	95%CI	
Age (≤50 vs. >50)	−1.32	0.717	3.385	0.066	0.267	0.0655	1.09							
Gender (Male vs. Female)	−1.243	0.902	1.9	0.168	0.288	0.0492	1.69							
AJCC stage (I/II vs. III/IV)	−1.676	0.871	3.705	**0.054**	0.187	0.034	1.031	−2.031	0.604	11.305	**<0.001**	0.131	0.0402	0.429
T stage (T1 vs. ≥T1)	−1.007	0.91	1.226	0.268	0.365	0.0614	2.172							
N stage (0 vs. ≥1)	0.164	0.607	0.0731	0.787	1.178	0.359	3.869							
M stage (0 vs. 1)	1.15	0.491	5.482	**0.019**	3.157	1.206	8.263	1.139	0.462	6.076	**0.014**	3.122	1.263	7.721
PDK1 expression (high vs. low)	−1.529	0.53	8.33	**0.004**	0.217	0.0767	0.612	−1.282	0.449	8.16	**0.004**	0.278	0.115	0.669

## Data Availability

The datasets used and analyzed in the current study are publicly accessible as indicated in the manuscript.

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
