# Peer review of "PDK1 Inhibitor BX795 Improves Cisplatin and Radio-Efficacy in Oral Squamous Cell Carcinoma by Downregulating the PDK1/CD47/Akt-Mediated Glycolysis Signaling Pathway"

_ijms, 2021, doi:10.3390/ijms222111492_

Round 1

Reviewer 1 Report

In this paper titled "PDK1 inhibitor BX795 improves cisplatin and radio-efficacy in oral squamous cell carcinoma by downregulating the PDK1/CD47/Akt-mediated glycolysis signalling pathway" the authors clearly outline how BX795 improves the effect of cisplatin and radio-efficacy in combating oral squamous cell carcinoma.

The manuscript is well designed and well presented and the authors have provided a clear mechanism. Overall the data presented may add value to the current set of information. 

The manuscript may be accepted for publication.

Author Response

Point-by-point responses to reviewer's comments:

Reviewer #1:

In this paper titled "PDK1 inhibitor BX795 improves cisplatin and radio-efficacy in oral squamous cell carcinoma by downregulating the PDK1/CD47/Akt-mediated glycolysis signalling pathway" the authors clearly outline how BX795 improves the effect of cisplatin and radio-efficacy in combating oral squamous cell carcinoma.

The manuscript is well designed and well-presented, and the authors have provided a clear mechanism. Overall, the data presented may add value to the current set of information.

The manuscript may be accepted for publication.

A: The reviewer's encouraging comment and appreciation is greatly appreciated.

Reviewer 2 Report

Shin Pai and colleagues report in their manuscript titled “PDK1 inhibitor BX795 improves cisplatin and radio-efficacy in oral squamous cell carcinoma by downregulating the PDK1/CD47/Akt-mediated glycolysis signalling pathway” some data indicating that expression level of PDK1is the important factor in Akt/CD47/LDHA signalling cascade and induced EMT in OSCC cells. Authors demonstrated the molecular mechanism of the interaction between PDK1and CD47 during tumorigenesis of OSCC. The main finding of  the present work is observation that novel PDK1 inhibitor - BX795 improves the effectiveness of chemo- and radiotherapy by regulating the expression of CD47/Akt/PDK3/LDHA in OSCC. The concept, the performance and the interpretation of the experiments are convincing. The methods used prove author’s expertise in molecular biology to perform this study.

Minor points:  

1.     I strongly recommend the unify the  correct format for human gene and protein symbols in the whole article. Human gene symbols are italicized.  (e.g., (PDPK1).
2.     Fig. 2 B- no description of units.
3.     Fig. 4 C, I can not find any information about the cisplatin concentration and BX795.
4.     Fig. 4E  - no description of units.
5.     I can not find any information about cell lines SAS and TW2.6 which were use in experiment: what passage were used?
6.     Fig. 5 I can not find any information about the cisplatin concentration and BX795.

Author Response

Point-by-point responses to reviewer's comments:

Dear Reviewer,

Coauthors and I very much appreciate the encouraging, critical, and constructive comments on this manuscript by the reviewer. The comments have been very thorough and useful in improving the manuscript. We strongly believe that the comments and suggestions have increased the scientific value of the revised manuscript by many folds. We have taken them fully into account in revision. We are submitting the corrected manuscript with the suggestion incorporated in the manuscript. The manuscript has been revised as per the comments given by the reviewer, and our responses to all the comments are as follows:

Reviewer #2:

Shin Pai and colleagues report in their manuscript titled “PDK1 inhibitor BX795 improves cisplatin and radio-efficacy in oral squamous cell carcinoma by downregulating the PDK1/CD47/Akt-mediated glycolysis signalling pathway” some data indicating that expression level of PDK1is the important factor in Akt/CD47/LDHA signalling cascade and induced EMT in OSCC cells. Authors demonstrated the molecular mechanism of the interaction between PDK1and CD47 during tumorigenesis of OSCC. The main finding of the present work is observation that novel PDK1 inhibitor - BX795 improves the effectiveness of chemo- and radiotherapy by regulating the expression of CD47/Akt/PDK3/LDHA in OSCC. The concept, the performance and the interpretation of the experiments are convincing. The methods used prove author’s expertise in molecular biology to perform this study.

A: We would like to thank the Reviewer for the thorough reading of our manuscript as well as the valuable comments and appreciation. We have followed the reviewer's comments thoroughly and added the extra information that was previously lacking on the figures and drug dosage used. We feel that they have further helped in strengthening our manuscript.

Minor points:  

Q1: Reviewer #2: I strongly recommend the unify the correct format for human gene and protein symbols in the whole article. Human gene symbols are italicized.  (e.g., (PDPK1).

A1: We thank the reviewer for the insightful comments. We have changed the format of gene symbols as per the suggestion. Please kindly refer to the main text of the attached manuscript.

Q2: Reviewer #2: Fig. 2 B- no description of units.

A2: The reviewer's insightful comments are greatly appreciated, in this revised manuscript we have added the description of Figure 2B units, kindly refer to the revised Figure 2B.

Q3: Reviewer #2: Fig. 4 C, I cannot find any information about the cisplatin concentration and BX795.

A3: We thank the reviewer for the comments and apologize for our oversight. We agree with the reviewer's comment, we also cross-checked observe the same, we have added the concentration of cisplatin and BX795 used. Please kindly refer to our revised Figure 4C.   

Q4: Reviewer #2: Fig. 4E - no description of units.

A4: We apologized for not showing the description of units. The reviewer's insightful comments are greatly appreciated, in this revised manuscript we have added the unit description of Figure 4E, kindly refer to the revised Figure 4E.

Q5: Reviewer #2: I cannot find any information about cell lines SAS and TW2.6 which were use in experiment: what passage were used?.

A5: We again highly appreciate the reviewers’ insightful and helpful comments on our manuscript and apologize for our oversight. We have added the cell lines SAS and TW2.6 passage used for this study. Please kindly refer to the main text of the manuscript, at page 18 and lines 505-506.

Q6: Reviewer #2: Fig. 5 I cannot find any information about the cisplatin concentration and BX795.

A6: We thank the reviewer for the comments and apologize for our oversight. We agree with the reviewer's comment, we also cross-checked observe the same, we have added the concentration of cisplatin and BX795 used. Please kindly refer to our revised Figure 5.   
